# Effectiveness of roadside alcohol testing in reducing fatal accidents and fatal drinking-driving accidents: A multi-city study in China

**Feng Li** *, **Xi Nie**

Department of Sociology, Faculty of Humanities, Southeast University, Nanjing, China

* phoneli@seu.edu.cn

## Abstract

This study examines the dynamic relationship between roadside alcohol check rates and traffic mortality across 248 cities in mainland China from 2014 to 2020. Using a dataset comprising 365,753 roadside check arrests, 227,896 traffic deaths, and 21,036 DUI-related fatalities, we applied both traditional time series analysis (Vector Autoregression and Impulse Response Functions) and machine learning techniques (XGBoost) to explore temporal and nonlinear patterns. The time series analysis revealed an initial positive association between enforcement intensity and mortality rates, likely reflecting reactive increases in checks following fatality rises. However, a longer-term beneficial effect emerged after approximately eight months, particularly pronounced in smaller cities. Machine learning models revealed that roadside check rates are more strongly associated with overall traffic mortality than with DUI-specific deaths. Enforcement was found to have a greater impact in smaller cities compared to larger ones. Notably, the patterns reveal diminishing returns in enforcement effectiveness, with the marginal benefits tapering off around 0.002 per 100,000 people per month in large cities, while remaining evident up to about 0.006 in small cities. These findings suggest that increasing roadside checks in low population density areas may lead to the most significant reductions in traffic fatalities at the national level. However, limitations include data exclusions due to non-disclosure and the inability to determine causal mechanisms. Overall, the study offers valuable insights for optimizing DUI enforcement strategies, highlighting the importance of tailored approaches based on city size and enforcement thresholds.

## Introduction

Road traffic injuries constitute a significant global health challenge, resulting in approximately 1.2 million fatalities each year and ranking as the foremost cause of unintentional injury deaths globally [1]. Among numerous risk factors, driving under the influence of alcohol (DUI) stands out as a critical contributor to traffic fatalities,

**Data availability statement:** All relevant data are within the paper and its Supporting Information files.

**Funding:** This work was supported by the National Social Science Fund of China (Grant No. 25BKX058) received by Dr. Feng Li.

**Competing interests:** The authors have declared that no competing interests exist.

accounting for between 5% and 35% of such incidents across different countries and regions [2].

To address the issue of DUI, 174 countries have enacted national drink-driving legislation, and 136 of these countries have established blood alcohol concentration (BAC) threshold limits [3]. These laws serve as a general deterrent to reduce DUI incidents and overall traffic fatalities [3]. The effectiveness of punishment as a general deterrent could be influenced by three key factors: certainty (the probability that an offense will be detected, apprehended, and punished), severity (the harshness of the punishment), and celerity (the promptness with which punishment is administered following the offense). Consequently, despite similar legal provisions against DUI, the deterrent effects vary significantly across different countries and regions [4].

The intensity of DUI enforcement plays a critical role in instilling fear of punishment among potential offenders, and are widely considered the most important factors in achieving a deterrent effect [4]. However, limited research has investigated the temporal impact of DUI enforcement intensity on traffic fatalities. To address this gap, the present study aims to examine the relationship between enforcement intensity and traffic fatality trends over time. The findings are expected to offer practical insights for law enforcement agencies, supporting more effective resource allocation and the development of targeted DUI enforcement strategies.

The literature review examines existing research on the deterrent effects of DUI legislation and law enforcement. While DUI laws are widely acknowledged as essential for reducing traffic accidents and fatalities, comparatively less attention has been given to the role of law enforcement in amplifying the effectiveness of these laws. This section begins by reviewing the impact of DUI legislation, followed by an analysis of various DUI law enforcement strategies, with particular emphasis on the temporal dynamics between enforcement intensity and its deterrent effects. The review concludes by identifying key gaps in the literature and outlining a research direction aimed at addressing these shortcomings.

### Examining the deterrence effective of DUI legislation

European countries and the USA were among the pioneers in DUI legislation [5]. For instance, in California, USA, the introduction of per se DUI laws in 1981 made it illegal to drive with a BAC of 0.10% or higher, irrespective of impairment [6]. To examine the effectiveness of DUI legislation, Rogers and Shoenig analyzed statewide monthly fatal and injury rates from 1979 to 1986 using an interrupted time-series analysis to compare the rates before and after the implementation of these laws [6]. Their findings indicated that the legislation served as a significant factor in reducing these rates.

In developing countries, the introduction of DUI legislation occurred later compared to the USA and European countries. For instance, Brazil enacted its first DUI-related law in 1997, making it illegal to drive with a BAC higher than 60 mg/100 ml [7]. In 2008, Brazil further amended the law to adopt a zero-tolerance policy for DUI, setting the legal BAC limit at 20 mg/100 ml [8]. However, while the mortality rates for pedestrians decreased, those for motorcyclists and cyclists increased [7]. Furthermore,

Volpe et al. found no significant evidence of a reduction in traffic-related mortality following DUI legislation in three major Brazilian cities: Belo Horizonte, Rio de Janeiro, and São Paulo [8].

The findings suggest that while the implementation of DUI legislation is essential, its effectiveness in reducing traffic-related fatalities largely depends on the level of enforcement. Legislation alone may be insufficient to generate a meaningful deterrent effect without consistent and visible enforcement efforts. In other words, when enforcement intensity is low, the perceived certainty of punishment among potential offenders decreases, thereby undermining the law's capacity to deter impaired driving and reduce traffic-related mortality.

### DUI legislation and law enforcement in China

The DUI legislation in China was established significantly later compared to that in the United States and Brazil. Specifically, China enacted its DUI legislation in May 2011, categorizing driving with a BAC exceeding 20 mg/100 ml as a civil offense and BAC exceeding 80 mg/100 ml as a criminal offense, specifically categorized as the crime of dangerous driving (危险驾驶罪) [3]. Moreover, in the event of a fatal traffic accident, if a driver's BAC level exceeds 20 mg/100 ml, the incident is classified as being caused by DUI and categorized as the crime of traffic violation (交通肇事罪) [9]. An interrupted time-series analysis was utilized to assess the deterrent effects of this DUI-related law. The results indicated that crash, mortality, and injury rates showed a gradual decline from 2011 to 2017, this suggests that the general deterrent effect of the legislation may have been effective [10].

Since the implementation of DUI legislation in China, the number of individuals convicted of dangerous driving has increased substantially, accounting for approximately one-quarter of all criminal cases nationwide between 2019 and 2020 [11]. However, according to traffic police data, annual traffic-related fatalities have remained relatively stable at around 60,000 deaths per year [9]. This suggests that, despite the sharp rise in criminal convictions driven by stricter DUI laws, these measures alone may have had limited impact on reducing overall traffic fatalities.

### The gap in prior research regarding methodological considerations

DUI enforcement consists of two principal stages: police roadside apprehensions and subsequent judicial proceedings [12]. Roadside apprehensions can heighten the perceived probability of arrest, and the proceeding conviction of crime can reinforce the perception that driving under the influence is unlawful. Both factors contribute to enhancing the certainty of deterrence. However, relative to the substantial body of research on the deterrent impact of legislation, there is a notable paucity of studies specifically addressing DUI law enforcement [13,14].

In the USA, Yao et al. employed mixed-effects regression models and found that enforcement intensity, particularly annual DUI arrest rates, uniquely and significantly predicts reductions in alcohol-impaired crash fatalities. Even after controlling for factors such as traffic funding and demographic variables, increased DUI enforcement remained strongly associated with declines in DUI-related deaths. These findings highlight that legislation alone is insufficient, that effective and visible enforcement is crucial for deterring drunk driving and reducing fatal crashes, especially in urban areas [14]. However, Elvik argued that the relationship between enforcement intensity and crash reduction is non-linear. Specifically, he noted that diminishing returns occur as enforcement levels increase. That is, additional enforcement yields progressively smaller reductions in traffic crashes [15]. Little is known about how this non-linear relationship varies across different city sizes or enforcement contexts, particularly in countries like China where empirical evidence remains scarce.

In China, Zheng highlighted regional disparities in law enforcement intensity, noting that economically developed areas tend to allocate more resources to DUI enforcement, leading to higher level of deterrence. In contrast, enforcement efforts are relatively weaker in some remote regions due to limited law enforcement capacity [11]. Li et al. observed that in economically thriving first-tier cities in China, the mortality rates were lower compared to other tiers of cities [9]. Bhalla et al. studied the prevalence of speeding and drinking driving in two Chinese cities as part of a mid-project evaluation of ongoing road safety interventions. They found that targeted enforcement and public awareness campaigns were associated

 

with a measurable reduction in these risky behaviors in the intervention city, while rates remained relatively unchanged in the control city [16]. Nevertheless, empirical studies examining the intensity of DUI law enforcement and its relationship with traffic mortality in China remain extremely limited. As a result, it is still unclear whether an increase in enforcement intensity could lead to a meaningful reduction in traffic-related deaths.

Previous studies have primarily relied on traditional time-series approaches, such as interrupted time series analysis, which can only compare changes in mortality trends between two time periods and are therefore typically applied to evaluate pre- and post-legislation effects. In contrast, mixed-effects regression models can accommodate hierarchical structures and control for observed confounders [14]; however, they generally impose a linearity assumption that may not adequately capture the complex relationship between enforcement intensity and traffic mortality. Moreover, traffic accident data often exhibit weak temporal dependence and non-stationary characteristics, making traditional time-series modeling challenging [17]. Therefore, it is necessary to complement conventional regression approaches with machine learning models that relax the linearity assumption, enabling a more flexible assessment of enforcement impacts while more effectively addressing confounding factors [18]. In addition, the use of SHapley Additive exPlanations (SHAP) values provides an interpretable framework to quantify the contribution of enforcement, among a set of potential confounders, to variations in mortality rates [19].

Therefore, to examine the temporal relationship between DUI law enforcement and traffic mortality, this study employs a combination of traditional time-series analysis and machine learning techniques to analyze the dynamic deterrent effects of DUI enforcement in China, aiming to provide actionable insights for refining enforcement strategies and enhancing traffic safety.

## Methods

### Data source

This study utilized voluntarily disclosed government documents, which are publicly accessible. Given the nature of these publicly available data sources, ethical review was not required. Data analysis was conducted using Python (version 3.12.4 on Windows). The data on roadside check and fatal traffic accidents used in this study were obtained from first-instance court judgment documents voluntarily disclosed by Chinese courts and available through the National Basic Science Data Center [20]. However, comprehensive disclosure only became possible after 2013, when a mandatory court file disclosure policy was implemented. Prior to that, very few traffic crime cases were publicly accessible. Furthermore, the implementation of the Data Security Law in 2021 significantly reduced the number of publicly disclosed court documents [21]. For these reasons, the study period is limited to traffic crime cases that occurred between 2014 and 2020, during which the availability of case data was most complete for epidemiological analysis.

In addition to traffic-related data, the study incorporates geographic, population, and socio-economic variables. Geographic information is based on the official 2019 administrative map of China (ID: GS(2019)1822). Population data were sourced from the sixth (2010) and seventh (2020) national censuses [22], while socio-economic indicators were obtained from municipal-level statistical bureaus.

### Text mining

Chinese court documents follow a standardized structure, typically beginning with a summary of the criminal behavior, followed by a legal analysis linking the behavior to its consequences, and concluding with the sentencing outcome. To extract key information from these documents, we employed rule-based keyword matching using Python's re package. The extracted dataset included information on the year and month of each traffic crime, the city in which the offense occurred, whether the offender was apprehended during a roadside DUI check, and the number of fatalities resulting from the incident. The code used for the data extraction process is available in the supplementary file S1 File (*supple_code.txt*).

After applying text-mining techniques, a total of 381,233 arrests resulting from roadside alcohol checks were identified during the study period. Additionally, 248,973 deaths caused by traffic violations were recorded, of which 23,111 were associated with DUI-related offenses.

## Data preparation

Among the 360 cities in mainland China, 76 cities reported no roadside check arrests or traffic-related mortalities during any year of the study period. In addition, 75 cities had populations of fewer than 1 million residents, according to the Seventh National Census. To ensure data stability and reduce the impact of low case frequency or small population size, these cities were excluded from the analysis. After this screening process, 248 cities remained for inclusion in the study (Fig 1). This exclusion accounted for only 4.1% of roadside check arrests (leaving 365,753 cases), 8.5% of total traffic-related mortalities (227,896 cases remaining), and 9.0% of DUI-related mortalities (21,036 cases remaining). As a result, the final dataset retained the vast majority of roadside checks and traffic-related fatalities that occurred in China during the study period.

A linear interpolation method was used to estimate annual population figures for each city between the 2010 and 2020 census years. Based on these estimates, roadside check rates and mortality rates were calculated per 100,000 population for each of the 248 selected cities on a monthly basis from 2014 to 2020. These rates served as the independent variable

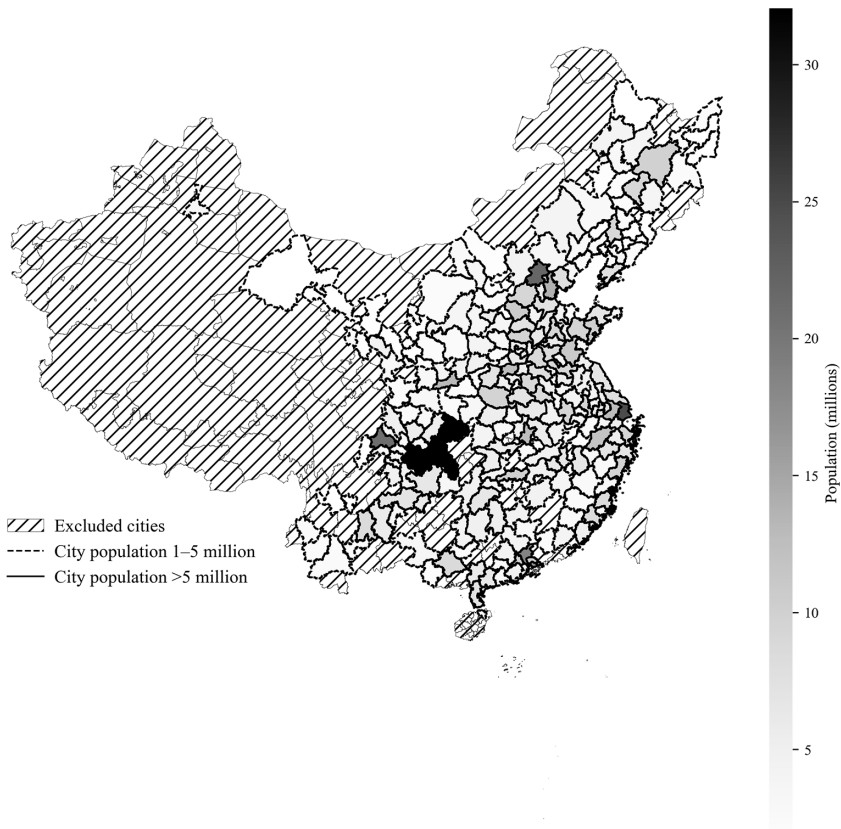

**Fig 1. Study regions and city population.** Downloaded from the Standard Map Service System of the Ministry of Natural Resources of the People's Republic of China (http://bzdt.ch.mnr.gov.cn/). The base map was freely provided for public use and modified by the authors to include population data from the national census.

(roadside check rate) and two dependent variables (overall traffic mortality rate and alcohol-related mortality rate) in the subsequent analysis. For months with zero reported cases, a minimal value (1e-6) was imputed to enable the application of classical time-series analytical methods and machine learning models.

A total of 4 socio-economic factors were acquired from city-level statistical bureaus and served as confounding variables, including GDP (100,000,000 RMB) per 100,000 people, road passenger volume (10,000) per 100,000 people, financial income of city government (10,000 RMB) per 100,000 people, and number of hospital beds per 100,000 people. Since these variables were originally available only as annual data, monthly values were obtained by linear interpolation between yearly observations to better match the monthly frequency of the dependent and independent variables. This interpolation assumes gradual changes within each year and is unlikely to introduce significant bias, as socio-economic indicators typically vary smoothly over time. Information on dependent, independent, and confounding variables can be found in the supplementary file S1 File (*supple_df.csv*).

## Vector autoregression and impulse response analysis

To analyze the dynamic relationship between roadside alcohol checks and mortality outcomes, we employed a Vector Autoregression (VAR) model separately for small (1–5 million population in 2020) and large cities (> 5 million population in 2020). To ensure the VAR analysis, we first conducted unit root tests to examine the stationarity of time series variables. Specifically, the Augmented Dickey-Fuller (ADF) test was applied separately to the monthly panel data of small and large cities. The VAR model captures the linear interdependencies among multiple time series variables. Specifically, we modeled the response of total traffic mortality and alcohol-related traffic mortality to shocks in roadside check rates over time. The analysis was conducted on monthly panel data aggregated by city size. For each city group (small or large), we estimate the following VAR model:

$$Y_t^{(g)} = A_1 Y_{t-1}^{(g)} + A_2 Y_{t-2}^{(g)} + \ldots + A_p Y_{t-p}^{(g)} + u_t^{(g)}$$

Where $Y_t^{(g)}$ is $\left[\binom{roadside\ check\ rate_t}{mortality\ rate_t}\right]$ or $\left[\binom{roadside\ check\ rate_t}{alochol\ mortality\ rate_t}\right]$, and $u_t^{(g)}$ denotes a vector of innovations. The optimal lag order $p$ was selected using the Akaike Information Criterion (AIC) and applied consistently across model estimation and Impulse Response Functions (IRFs) analysis. To ensure robustness, additional estimations were conducted using alternative lag lengths ($p = 4\ and\ 8$).

We computed the IRFs to trace the dynamic impact of a one-time shock to roadside alcohol check rates on mortality outcomes over the following 1–8 months. Formally, the IRF at horizon $h$ is defined as:

$$IRF(h) = \frac{\partial Y_{t+h}}{\partial \epsilon_t}$$

Where $\epsilon_t$ is a one-unit innovation (shock) in the roadside check variable. We present IRFs with 95% confidence intervals computed using standard errors from the VAR estimates. Separate models were estimated for small and large cities to assess heterogeneous effects across different urban sizes.

## Gradient Boosted Regression for marginal effect estimation

To account for confounding effects that traditional time-series methods cannot adequately address, we employed Gradient Boosted Regression Trees (GBRT) implemented via the XGBoost library. Before finalizing the model, we compared three candidate algorithms, namely Random Forest, Support Vector Regression, and XGBoost. A grid search procedure was used to identify the optimal set of hyperparameters for each model. Based on both predictive performance and interpretability considerations, XGBoost was selected as the final model because it provided an appropriate balance between coefficient of determination ($R^2$) and root mean squared error (RMSE).

This machine learning approach is well suited to capturing complex, nonlinear relationships and interaction effects among predictors without requiring prior assumptions about their functional form. To further evaluate the contribution and impact pattern of each feature in the model, we utilized both SHAP values and marginal effect analysis. SHAP values offer a unified measure of feature importance and directionality at both global and local levels, while marginal effect curves illustrate the predicted change in the outcome variable as a function of a specific predictor, holding all other variables constant. This complementary analytical framework enables us to identify not only which variables are most influential but also how they shape predicted mortality outcomes.

Separate models were trained for small and large cities, using the same dependent variables as in the VAR models: mortality rate and DUI-related mortality rate. The predictors included the roadside check rate and its lagged values (Lag0, Lag4, Lag8), along with four confounding variables. To account for seasonal effects, monthly seasonality features were created using sine and cosine transformations of the month variable. An XGBoost Regressor (100 estimators, max depth=4, learning rate=0.1) was employed to model the relationships between predictors and each outcome. For each combination of city group and outcome, 3-fold cross-validation was conducted by randomly splitting the data into three subsets; in each fold, two subsets were used for training and one for testing. Model performance was assessed using the $R^2$ and RMSE, averaged across the three folds. Feature importance was interpreted through SHAP values, with the mean absolute SHAP value quantifying each feature's overall impact, and the mean SHAP value indicating the directionality of its effect.

To visualize the marginal effect of roadside check intensity, we used partial dependence plots. For each lagged roadside check variable, we generated a sequence of hypothetical values across its observed range (5th to 95th percentile) while holding other variables constant at their mean values. The predicted outcomes were then plotted to depict how the mortality rate changes with varying roadside check intensity.

Formally, the partial dependence function for a single feature $x_s$ is defined as:

$$\hat{f}_s(x_s) = \frac{1}{n} \sum_{i=1}^{n} \hat{f}_s(x_s, X_{C_i})$$

where $\hat{f}$ is the trained model, $X_{C_i}$ are the fixed values of the other features (controls), and $x_s$ is the feature of interest (roadside check rate with different lags). Separate plots were generated for each outcome and city size group to illustrate potential heterogeneity in the effects. Lagged variables were used to explore delayed policy impacts.

## Results

### Roadside check rates and mortality rates by city size

Fig 2 illustrates the trends of three variables over the entire study period. In both small and large cities, the roadside check rate shows a marked increase starting in September 2019, followed by a sharp surge. However, the rate declined significantly in January and February 2020 due to the nationwide COVID-19 lockdown. After reopening in March 2020, roadside checks rebounded and have remained at elevated levels since. The overall mortality rate follows a similar pattern to the roadside check rate, while the alcohol-related mortality rate remains relatively stable throughout the period.

### IRF analysis

All time series passed the ADF test for stationarity, allowing for the direct application of the VAR model followed by IRF analysis. Fig 3 presents the IRF results. Regarding overall mortality rates, in small cities, the IRF reveals an initial positive response that peaks around the first month, indicating a short-term increase in mortality rates associated with roadside checks. This effect gradually diminishes and becomes negative by approximately the sixth month, suggesting a subsequent decline in mortality rates. The negative effect continues to intensify, reaching its lowest point by the eighth month.

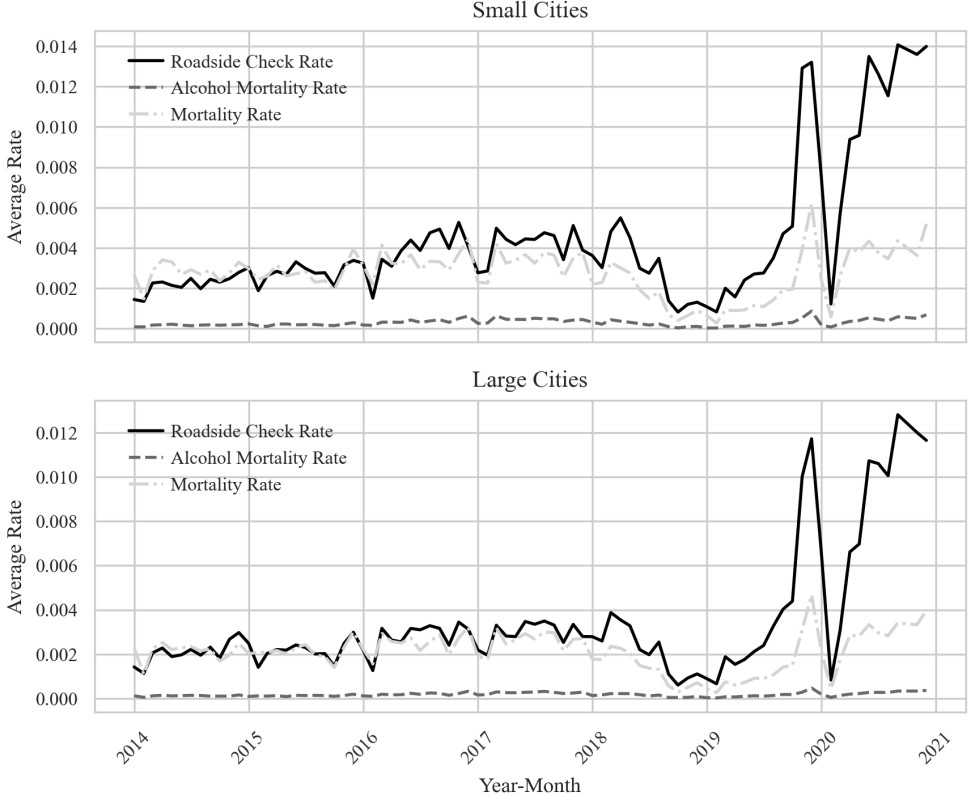

**Fig. 2. Trends in roadside check rates, mortality rates, and DUI-related mortality rates by city size (2014-2020).**

In large cities, a similar response pattern emerges, with an initial positive peak near the first month; however, the effect is less pronounced and more variable, showing fluctuations throughout the lag period. The response concludes with a negative effect by the eighth month, though this decline is less steep than in small cities.

For alcohol-related mortality rates, the IRF for small cities shows a strong initial positive response peaking around the second month, indicating a significant short-term increase in alcohol-related deaths linked to roadside checks. This response fluctuates thereafter, with a notable dip near the sixth month, followed by a slight recovery toward the end of the period. By the eighth month, the effect turns negative, reflecting a decrease in alcohol-related mortality rates. In large cities, the response is more muted, with a less pronounced initial peak and greater variability throughout the lag period. Similar to small cities, the response ends negatively by the eighth month, indicating a reduction in alcohol-related mortality rates. To test robustness, the VAR models were re-estimated using lag orders of 4 and 8. The impulse response functions exhibit highly consistent patterns across specifications. The peak responses occur around the first month in all cases, and the magnitude and direction of the effects change only marginally. These results confirm that the estimated IRFs are robust to alternative lag selections.

Overall, these findings suggest that roadside checks have differential impacts on mortality and alcohol-related mortality rates depending on city size, characterized by initial increases followed by decreases or stabilization in both outcomes. The 95% confidence intervals indicate some uncertainty in the estimates, particularly for large cities.

## SHAP feature importance with direction

Fig 4 presents the model performance and feature importance for predicting mortality rate and alcohol-related mortality rate using XGBoost regression, stratified by city size. For mortality rate, the model achieved an $R^2$ of 0.28 in small cities

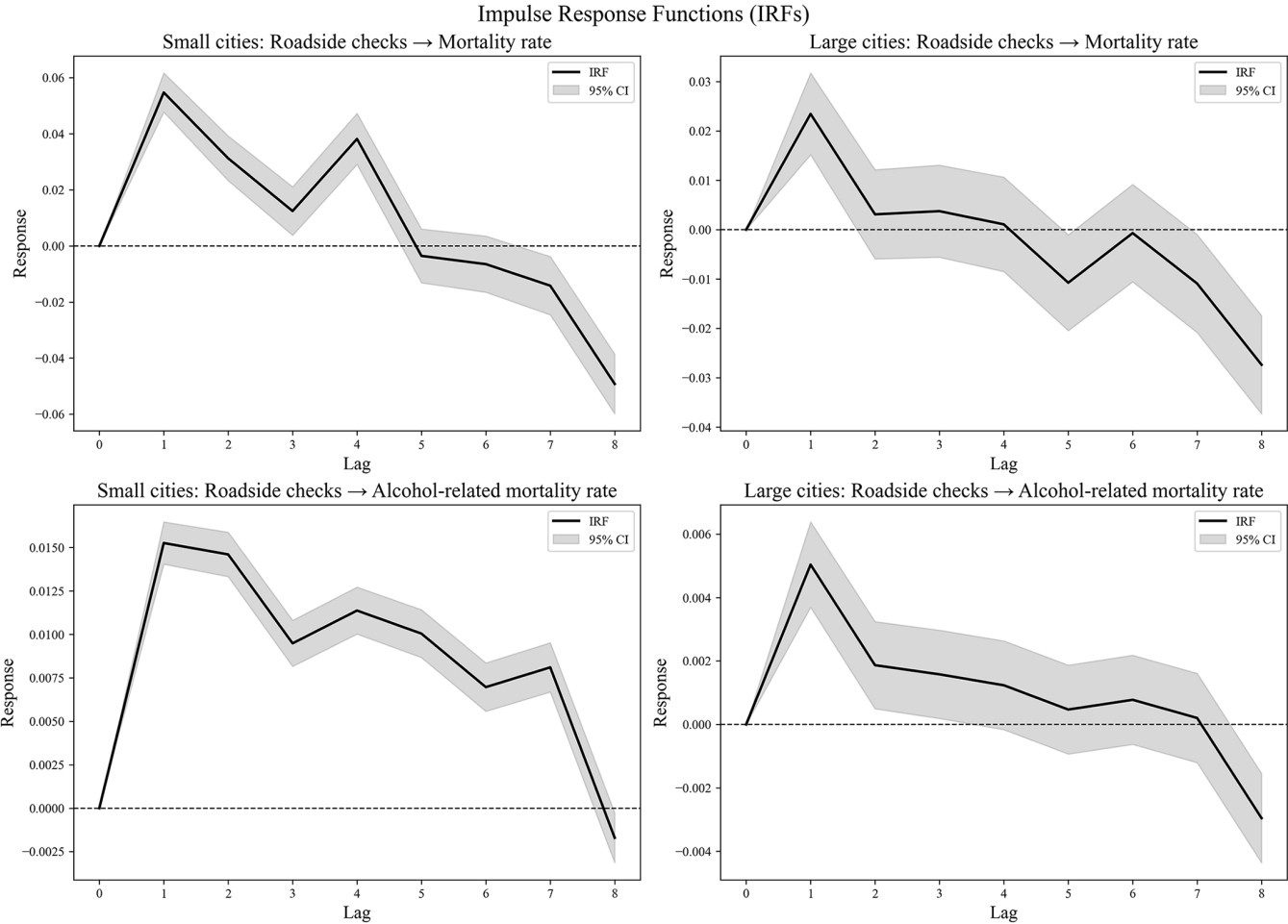

**Fig 3. The IRFs for the effects of roadside checks on mortality outcomes.**

and 0.42 in large cities, indicating a moderate fit. In contrast, models predicting alcohol-related mortality rate showed lower predictive power, with R² values of 0.06 and 0.07 for small and large cities, respectively.

Regarding overall mortality rates, roadside check rates and their lagged terms were consistently among the most important predictors across both city groups. In small cities, the current roadside check rate had the highest importance, substantially exceeding that of other factors. Conversely, in large cities, variables such as GDP per 100,000 people and hospital beds per 100,000 people demonstrated significant negative contributions. For both city groups, the 8-month lagged roadside check rate exhibited negative effects, suggesting a long-term reduction in mortality. Additionally, the cosine month term contributed positively, while the sine month term contributed negatively in both models, indicating that mortality rates tend to be higher during the winter months.

For alcohol-related mortality rates, roadside check rates (lag 0) were more important in small cities and exhibited a negative effect, implying a protective influence. In large cities, confounding variables such as financial income and passenger volume carried greater weight than in small cities, suggesting that the impact of roadside checks may be more pronounced in smaller urban cities. Furthermore, in both models of alcohol-related mortality, the cosine and sine terms for month demonstrated negative and positive associations, respectively, suggesting that alcohol-related mortality rates reach their peak during the summer months.

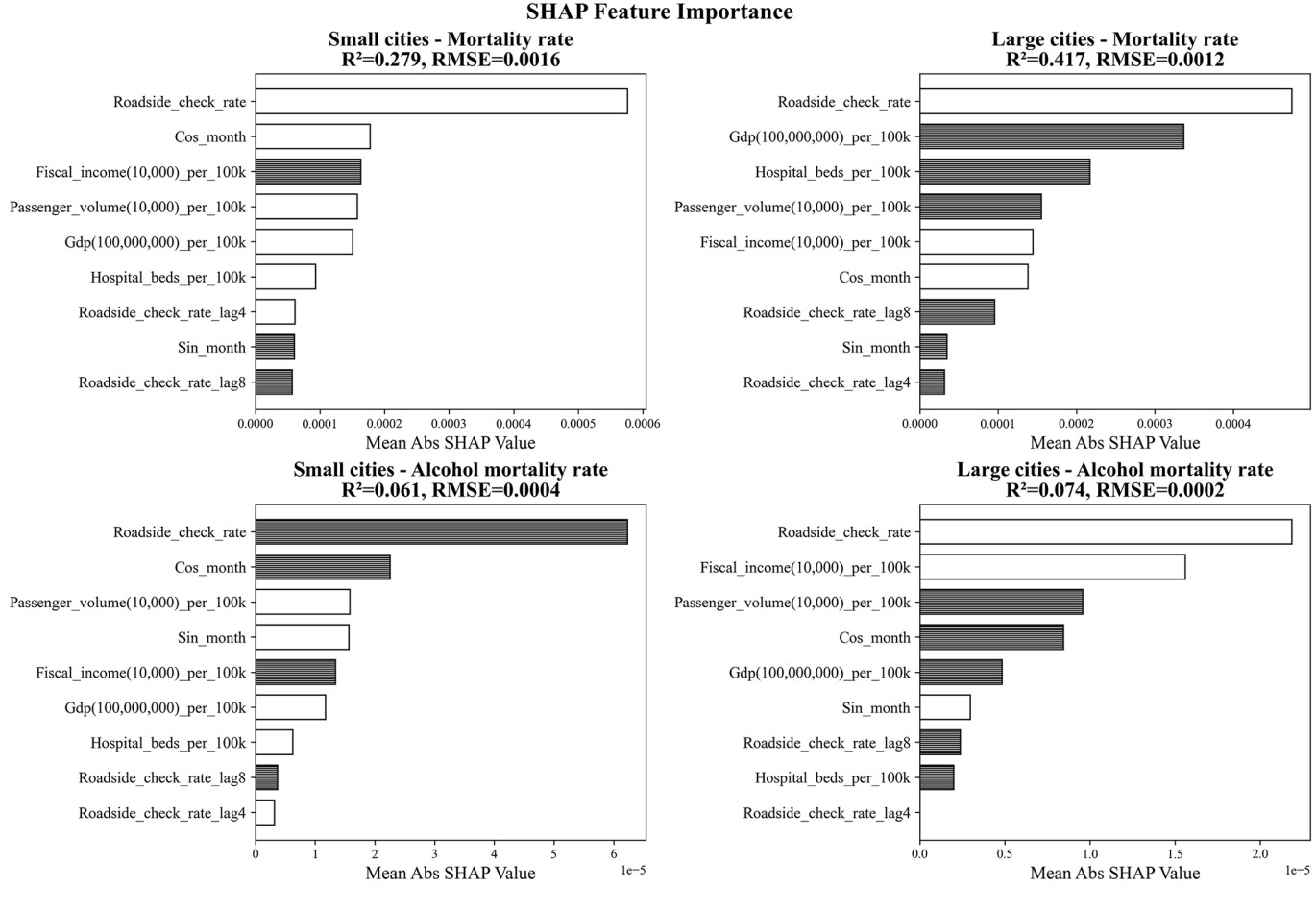

**Fig 4. Contribution analysis for mortality rate prediction using SHAP values.**

## Marginal effects of roadside check rate on mortality outcomes

Fig 5 illustrates the marginal effects of roadside check rates (lags 0, 4, and 8) on both overall mortality and alcohol-related mortality in small and large cities. At lag 0, mortality rates increase stepwise with higher roadside check rates in both city types, indicating a positive association. However, by lag 8, the relationship turns negative for most outcomes, except for alcohol-related mortality in small cities. Notably, the patterns indicate a diminishing returns effect on overall mortality rates, in large cities, where the marginal benefit of increasing roadside check rates levels off around 0.002. In contrast, small cities show a higher threshold, with the marginal effect persisting up to approximately 0.006 before stabilizing. These findings suggest that higher levels of roadside enforcement may be more effective in smaller cities.

## Discussion

This study utilized data from 365,753 roadside check arrests, 227,896 traffic deaths, and 21,036 DUI-related deaths across 248 cities in China between 2014 and 2020. On average, these figures correspond to approximately 52,251 individuals convicted annually of drunk driving, 32,556 people dying from traffic-related incidents, and 3,005 deaths specifically attributed to DUI offenses. It is important to note that court system data may underreport traffic-related mortalities

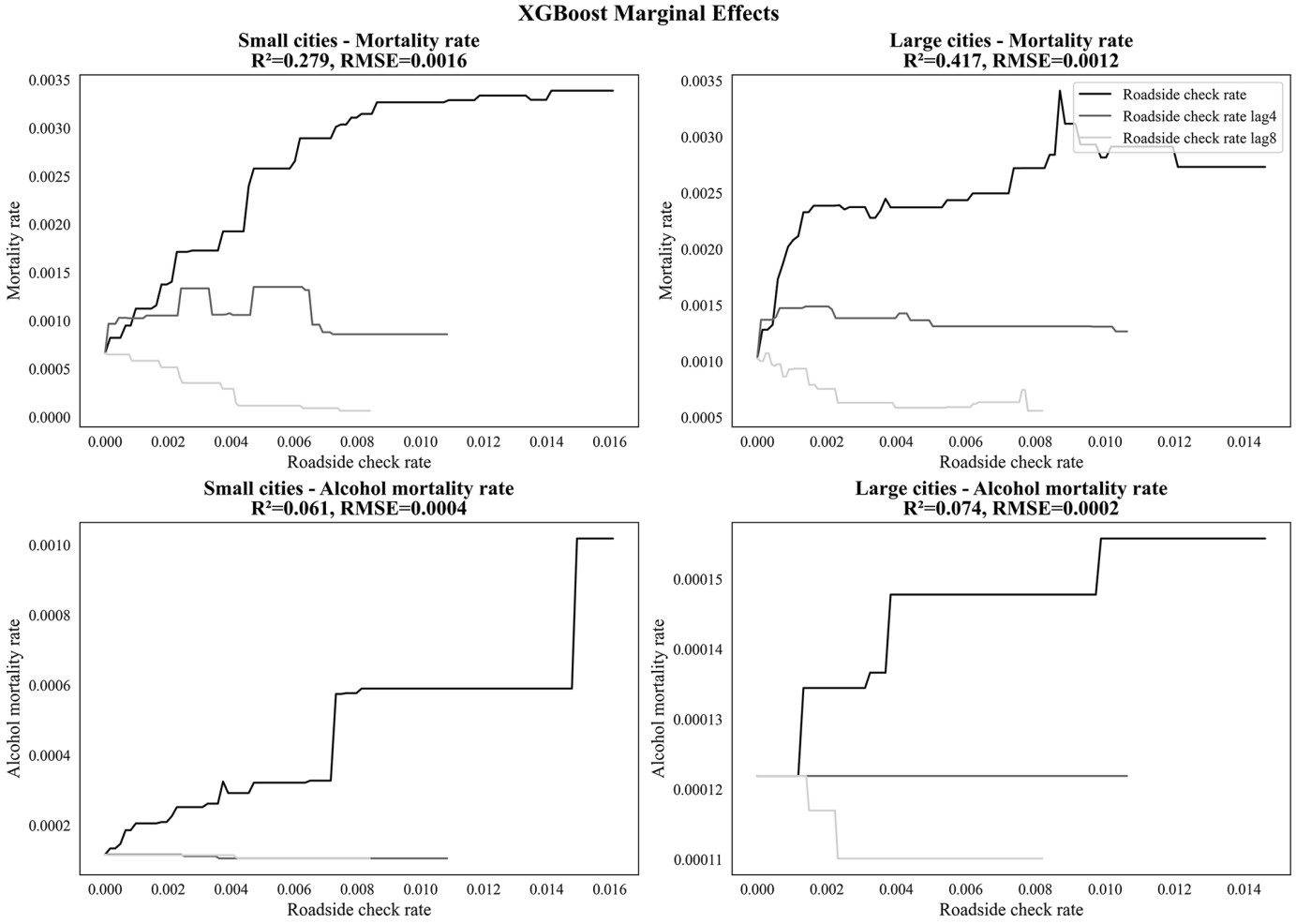

**Fig 5. Estimated marginal effects of roadside check rates (lags 0, 4, and 8 months) on mortality rates.**

in China, as court records typically include only cases in which the victim died at the scene and the offender survived. In contrast, the health department is able to capture fatalities that occur after prolonged hospitalization, as well as deaths of both victims and offenders [23]. Despite this potential underreporting, the number of cases retrieved from court documents remains substantial, underscoring the importance of investigating whether increased frequency of roadside checks contributes to a reduction in traffic-related fatalities.

The results provide a comprehensive analysis of the dynamic effects of roadside alcohol check rates on traffic mortality and alcohol-related mortality. By combining traditional time series methods with machine learning techniques, we captured both linear temporal dependencies and complex nonlinear relationships while effectively controlling for confounding factors. Specifically, the VAR model identifies temporal dynamics and lagged associations, whereas XGBoost uncovers nonlinear and marginal patterns across city groups. This dual strategy enhances the robustness of policy evaluation by addressing both time-dependent relationships and functional heterogeneity in the data.

The IRF analysis revealed an initial significant positive association between roadside check rates and mortality outcomes in the first few months. This finding may indicate that increased roadside checks are reactive to rising mortality rates, which is, enforcement intensity increases in response to higher fatalities to curb the trend. Consequently, a

longer-term negative effect, particularly evident at the 8-month lag, was observed, reflecting a delayed but beneficial reduction in fatalities due to enforcement efforts. Notably, the short-term increase followed by a decline was more pronounced in small cities, with the 95% confidence intervals of the IRFs more often reaching statistical significance in these areas. These results suggest that roadside alcohol checks may be more effective in reducing mortalities in smaller cities compared to larger urban centers.

Across all traffic crime records nationwide, 4.1% of roadside check arrests, 8.5% of mortalities, and 9.0% of DUI-related mortalities were excluded from the analysis, with most of these cases occurring in cities with populations under 1 million. This suggests that in cities with low population density, the frequency of roadside checks is relatively low, while mortality rates, particularly alcohol-related mortalities could be higher than in the cities included in this study. Therefore, increasing the frequency of roadside checks in these low-density cities may offer the great potential for reducing traffic fatalities.

The machine learning model results further indicated that roadside check rates are more closely associated with overall mortality rates than with DUI-specific mortality rates. This implies that the frequency of roadside checks arranged by municipal governments may be more related to the general likelihood of fatal crashes rather than DUI-related incidents specifically. In small cities, the most influential predictor was the current roadside check rate (lag 0), far surpassing other factors. The second most influential factor was the cosine month term, while socio-economic variables played only a minor role in mortality rates. Consequently, both machine learning and traditional time series analyses converge on the finding that law enforcement has a stronger impact on traffic mortalities in smaller cities compared to larger ones.

The findings support the observation of diminishing returns in enforcement levels [15]. Marginal effect analysis revealed a pattern of diminishing returns in large cities, where reductions in mortality began to stabilize once roadside check rates reached approximately 0.002 per 100,000 people per month. In contrast, small cities showed a higher threshold effect, with marginal impacts persisting up to around 0.006 before leveling off. This disparity may reflect differences in enforcement efficiency: in large cities, increasing roadside check and arrest rates beyond 0.002 per month may yield little additional benefit in reducing traffic fatalities. Conversely, in small cities, higher monthly rates up to about 0.006 are needed to effectively curb potentially elevated traffic mortality risks. Beyond these thresholds, further increases appear to have little additional effect in either setting.

Consistent with Zheng's study [11], our results emphasize the importance of increasing roadside check frequency in smaller cities, where such measures have proven effective in reducing fatalities. However, there appears to be an upper threshold of about 0.006 in arrest rates, beyond which the marginal gains in fatality reduction diminish. Unlike Zheng's findings, our study suggests that roadside checks are less effective in larger cities, where the marginal returns of increased enforcement intensity are notably lower. This indicates that as city populations grow, the benefits of intensified enforcement efforts decline.

## Limitations

First, this study is based on data from China, and the results may not be directly generalizable to other countries or regions. Significant differences in enforcement culture, social norms, and DUI laws across countries could impact the generalizability of policy effects [8]. Second, other traffic safety measures or policies, such as speed limit regulations and traffic safety education, may interact with DUI enforcement and influence the study's findings. This study was unable to fully isolate the effects of these factors, which could lead to an overestimation or underestimation of the results. Furthermore, the impact of total vehicle miles traveled on traffic accidents and various traffic violations was not considered, as data was unavailable. Third, due to the non-disclosure of cases involving state secrets or juvenile offenders, this study was unable to include these subsets of data in the analysis. Specifically, traffic accident cases may involve juvenile offenders, but the lack of relevant data limits the generalizability and completeness of the findings, thereby affecting the accurate assessment of overall trends in traffic violations. Lastly, neither classical time series models nor machine learning methods were able to identify a causal pathway linking roadside checks to fatality rates. Therefore, although this study demonstrated a

close temporal association between DUI law enforcement and traffic fatality rates, the underlying mechanisms of this interaction remain unestablished.

## Conclusions

This study highlights the complex relationship between roadside alcohol check rates and traffic fatalities across Chinese cities of varying sizes. Findings indicate that increased enforcement is associated with short-term rises followed by longer-term reductions in mortality, with greater effectiveness observed in smaller cities. The evidence also points to diminishing returns at higher enforcement levels, especially in larger urban areas. While the study provides valuable insights for policy and enforcement strategies, further research is needed to clarify causal mechanisms and to explore the impact of additional traffic safety measures.

Future studies should merge court records with hospital trauma registries, police patrol logs, and insurance telematics to construct kilometer-based enforcement measures. They could then exploit staggered municipal rollouts or randomized checkpoints as instrumental variables or Difference-in-Differences (DID) shocks to verify whether the observed eight-month mortality reduction represents a true causal deterrent effect rather than reactive policing or unobserved confounding shocks.

## Supporting information

**S1 File. Supplementary material covering text-mining procedures and datasets used in statistical analysis.**
(ZIP)

## Author contributions

**Conceptualization:** Feng Li, Xi Nie.

**Data curation:** Feng Li, Xi Nie.

**Formal analysis:** Xi Nie.

**Writing – original draft:** Feng Li, Xi Nie.

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
