## [Decision Letter · Decision Letter 0]

28 Oct 2025

Dear Dr. Li,

Thank you for submitting your manuscript to PLOS ONE. After careful consideration, we feel that it has merit but does not fully meet PLOS ONE’s publication criteria as it currently stands. Therefore, we invite you to submit a revised version of the manuscript that addresses the points raised during the review process.

Please try to improve your paper and respond to all the reviewers' comments.

We look forward to receiving your revised manuscript.

Kind regards,

Quan Yuan, Ph.D.

Academic Editor

PLOS ONE

2. We note that Figure 1 in your submission contain [map/satellite] images which may be copyrighted. All PLOS content is published under the Creative Commons Attribution License (CC BY 4.0), which means that the manuscript, images, and Supporting Information files will be freely available online, and any third party is permitted to access, download, copy, distribute, and use these materials in any way, even commercially, with proper attribution. For these reasons, we cannot publish previously copyrighted maps or satellite images created using proprietary data, such as Google software (Google Maps, Street View, and Earth). For more information, see our copyright guidelines: http://journals.plos.org/plosone/s/licenses-and-copyright.

 1. You may seek permission from the original copyright holder of Figure 1 Publish the content specifically under the CC BY 4.0 license. 

Additional Editor Comments (if provided):

Reviewers' comments:

Reviewer's Responses to Questions

**Comments to the Author**

1. Is the manuscript technically sound, and do the data support the conclusions?

Reviewer #1: Yes

Reviewer #2: Yes

2. Has the statistical analysis been performed appropriately and rigorously?

Reviewer #1: Yes

Reviewer #2: Yes

3. Have the authors made all data underlying the findings in their manuscript fully available?

Reviewer #1: Yes

Reviewer #2: No

4. Is the manuscript presented in an intelligible fashion and written in standard English?

Reviewer #1: Yes

Reviewer #2: Yes

Reviewer #1: In the manuscript titled “Effectiveness of Roadside Alcohol Testing in Reducing Fatal Accidents and Fatal Drinking-Driving Accidents: A Multi-City Study in China”, the authors analyzed a large-scale dataset (248 cities, 2014–2020) and applied both VAR/IRF models and XGBoost with SHAP analysis to investigate the dynamic effects of roadside alcohol testing on traffic mortality. This study contains interesting findings and provides valuable evidence for understanding the deterrent effects of DUI enforcement, especially the heterogeneous effects between small and large cities.

However, the manuscript also presents some weaknesses in terms of structure, literature review depth, and data/method transparency. Therefore, major revision has to be done before this manuscript could be accepted for publication in PLOS ONE.

1. The transition between literature review and research gap is not sufficiently clear. The authors should streamline this part and highlight the novelty of their study more explicitly.

2. The review cites many studies, but some are descriptive rather than analytical. Please strengthen the discussion with recent works on machine learning applications in traffic safety and international comparisons of enforcement strategies. I suggest the author integrate these articles:Sight distance analysis of vehicles with driving automation on horizontal curves of as-built highway tunnels; A system theory based accident analysis model: STAMP-fuzzy DEMATEL; Temporal heterogeneity in traffic crash delays: causal inference from multi-scale time factors and sample-wise structural decomposition.So, I believe this research will be greatly improved.

3. The reliance on court judgment documents introduces potential selection bias (e.g., only cases with on-site deaths and surviving offenders are included). This limitation should be emphasized more clearly in the methodology and discussion.

4. Only XGBoost was employed in the machine learning part. Please justify why no comparison with other algorithms (e.g., Random Forest, LSTM) was attempted. This would enhance methodological robustness.

5. Some expressions could be more precise. For example, “mortality plateau once roadside check rates reach approximately 0.002” could be rephrased in more formal academic language.

6. Some references are incomplete or not in standard PLOS ONE format (e.g., [7]). Please revise and ensure DOI information is provided where available.

7. Please add a section on future research directions, such as integrating multi-source data (e.g., hospital, police, insurance) or applying causal inference methods (DID, IV).

Reviewer #2: This manuscript presents a valuable multi-city study in China investigating the effectiveness of roadside alcohol testing in reducing traffic fatalities, using a combination of Vector Autoregression (VAR) for temporal dynamics and XGBoost for nonlinear pattern analysis. The topic of this paper is very interesting. However, several revisions are needed to strengthen the manuscript’s rigor, clarity, and compliance with PLOS ONE standards.

1. Inconsistencies and gaps mar the reference section. First, Jomar et al. (2018) is labeled "Unpublished or source unspecified," which violates PLOS ONE’s preference for peer-reviewed or formally archived sources.

2. Some Chinese reference appears.

3. Though the study claims "all data are fully available without restriction," the Data Availability Statement lacks actionable details.

4. Several figure-related issues hinder interpretability and are very vague.

5.The study’s conclusions would benefit from additional robustness checks. For the VAR model, only AIC is used to select lag order—no sensitivity analysis is provided to confirm if results hold with different lags. For XGBoost, parameters are not tuned.

**Do you want your identity to be public for this peer review?** For information about this choice, including consent withdrawal, please see our Privacy Policy

Reviewer #1: No

Reviewer #2: No

---

## [Author Response · Author response to Decision Letter 1]

3 Nov 2025

PONE-D-25-37604

Effectiveness of Roadside Alcohol Testing in Reducing Fatal Accidents and Fatal Drinking-Driving Accidents: A Multi-City Study in China

PLOS ONE

Reply

All the required documents are ready for submission. The funding source has been included in the cover letter.

2. We note that Figure 1 in your submission contain [map/satellite] images which may be copyrighted. All PLOS content is published under the Creative Commons Attribution License (CC BY 4.0), which means that the manuscript, images, and Supporting Information files will be freely available online, and any third party is permitted to access, download, copy, distribute, and use these materials in any way, even commercially, with proper attribution. For these reasons, we cannot publish previously copyrighted maps or satellite images created using proprietary data, such as Google software (Google Maps, Street View, and Earth). For more information, see our copyright guidelines: http://journals.plos.org/plosone/s/licenses-and-copyright.

1. You may seek permission from the original copyright holder of Figure 1 Publish the content specifically under the CC BY 4.0 license.

Reply

Thank you for your notice regarding the potential copyright issue for Figure 1.

We confirm that the base map in Figure 1 was obtained from the Standard Map Service System of the Ministry of Natural Resources of the People's Republic of China, which is an official and publicly available government data source. According to the website,

“标准地图依据中国和世界各国国界线画法标准编制而成可用于新闻宣传用图、书刊报纸插图、广告展示背景图、工艺品设计底图等也可作为编制公开版地图的参考底图。”

Which can be translated as “The standard maps are compiled based on the drawing standards of national boundaries of China and other countries around the world. They can be used for news and publicity maps, illustrations in books and newspapers, background images for advertisements and exhibitions, design bases for handicrafts, and as reference base maps for producing publicly released maps.”

The dataset is openly accessible through the following official website:

http://bzdt.ch.mnr.gov.cn/download.html?searchText=GS(2019)1822%EF%BC%89

We used this publicly available government map as the base layer and overlaid population data derived from the national census to produce Figure 1. The final figure is an original composite created by the authors, not a reproduction of any copyrighted or proprietary map (such as Google Maps, Street View, or Earth).

Therefore, there is no copyright infringement concern, and the figure fully complies with the CC BY 4.0 license requirements for open-access publication.

In the revised manuscript, we have clarified this in the figure caption as follows:

Fig. 1. Study regions and city population. Downloaded from the Standard Map Service System of the Ministry of Natural Resources of the People's Republic of China (http://bzdt.ch.mnr.gov.cn/). The base map was freely provided for public use and modified by the authors to include population data from the national census.

Additional Editor Comments (if provided):

Reviewers' comments:

Reviewer's Responses to Questions

Comments to the Author

1. Is the manuscript technically sound, and do the data support the conclusions?

Reviewer #1: Yes

Reviewer #2: Yes

2. Has the statistical analysis been performed appropriately and rigorously?

Reviewer #1: Yes

Reviewer #2: Yes

3. Have the authors made all data underlying the findings in their manuscript fully available?

Reviewer #1: Yes

Reviewer #2: No

Reply

The original data can be accessed from the following public source:

Hu D, Chen X, Liu L. (2024). Full judgment document data for intelligent trial assistance [面向智能审判辅助的全量裁判文书数据]. National Big Data Service Platform [全国数据服务平台]. Available from: https://www.nbsdc.cn/general/dataDetail?id=66606e34195d266d328f3ca0&type=1

In addition, the codes used for data mining and the datasets applied in the VAR/IRF models and XGBoost with SHAP analyses have been included as supplementary files with this submission. These materials ensure that all analyses can be replicated and verified in accordance with PLOS ONE’s data availability policy.

4. Is the manuscript presented in an intelligible fashion and written in standard English?

Reviewer #1: Yes

Reviewer #2: Yes

5. Review Comments to the Author

Reviewer #1: In the manuscript titled “Effectiveness of Roadside Alcohol Testing in Reducing Fatal Accidents and Fatal Drinking-Driving Accidents: A Multi-City Study in China”, the authors analyzed a large-scale dataset (248 cities, 2014–2020) and applied both VAR/IRF models and XGBoost with SHAP analysis to investigate the dynamic effects of roadside alcohol testing on traffic mortality. This study contains interesting findings and provides valuable evidence for understanding the deterrent effects of DUI enforcement, especially the heterogeneous effects between small and large cities.

However, the manuscript also presents some weaknesses in terms of structure, literature review depth, and data/method transparency. Therefore, major revision has to be done before this manuscript could be accepted for publication in PLOS ONE.

1. The transition between literature review and research gap is not sufficiently clear. The authors should streamline this part and highlight the novelty of their study more explicitly.

2. The review cites many studies, but some are descriptive rather than analytical. Please strengthen the discussion with recent works on machine learning applications in traffic safety and international comparisons of enforcement strategies. I suggest the author integrate these articles:Sight distance analysis of vehicles with driving automation on horizontal curves of as-built highway tunnels; A system theory based accident analysis model: STAMP-fuzzy DEMATEL; Temporal heterogeneity in traffic crash delays: causal inference from multi-scale time factors and sample-wise structural decomposition. So, I believe this research will be greatly improved.

Reply

To add the transition between literature review and research gap, we have read the three articles and added a new paragraph to increase the smooth transition:

“Previous studies have primarily relied on traditional time-series approaches, such as interrupted time series analysis, which can only compare changes in mortality trends between two time periods and are therefore typically applied to evaluate pre- and post-legislation effects. In contrast, mixed-effects regression models can accommodate hierarchical structures and control for observed confounders [14]; however, they generally impose a linearity assumption that may not adequately capture the complex relationship between enforcement intensity and traffic mortality. Moreover, traffic accident data often exhibit weak temporal dependence and non-stationary characteristics, making traditional time-series modeling challenging [17]. Therefore, it is necessary to complement conventional regression approaches with machine learning models that relax the linearity assumption, enabling a more flexible assessment of enforcement impacts while more effectively addressing confounding factors [18]. In addition, the use of SHapley Additive exPlanations (SHAP) values provides an interpretable framework to quantify the contribution of enforcement, among a set of potential confounders, to variations in mortality rates [19].”

3. The reliance on court judgment documents introduces potential selection bias (e.g., only cases with on-site deaths and surviving offenders are included). This limitation should be emphasized more clearly in the methodology and discussion.

Reply

In the first paragraph of the Discussion, we have stated that:

“It is important to note that court system data may underreport traffic-related mortalities in China, as court records typically include only cases in which the victim died at the scene and the offender survived. In contrast, the health department is able to capture fatalities that occur after prolonged hospitalization, as well as deaths of both victims and offenders [23]. Despite this potential underreporting, the number of cases retrieved from court documents remains substantial, underscoring the importance of investigating whether increased frequency of roadside checks contributes to a reduction in traffic-related fatalities.”

4. Only XGBoost was employed in the machine learning part. Please justify why no comparison with other algorithms (e.g., Random Forest, LSTM) was attempted. This would enhance methodological robustness.

Reply

We appreciate the reviewer’s insightful comment. The three papers mentioned indeed focused mainly on engineering applications, where machine learning models are optimized for predictive accuracy to enhance traffic or engineering safety. In contrast, the current study emphasizes the policy dimension. Specifically, the effectiveness of law enforcement policies. Therefore, our modeling objective differs from that of engineering-oriented research.

Unlike engineering studies that pursue highly precise predictive performance, the mathematical modeling of law enforcement policy does not necessarily benefit from such optimization. Even if a model achieves the best performance on the current dataset, it may not generalize well under different political systems and cultural contexts, resulting in potential overfitting. Consequently, in this study we selected the XGBoost model from three candidate algorithms through a grid search for optimal hyperparameters, without exhaustively testing all machine learning methods or tuning a larger set of parameters. This decision reflects our intention to balance model interpretability and generalizability rather than to over-optimize performance on a single dataset.

In the Method part, we added:

“To account for confounding effects that traditional time-series methods cannot adequately address, we employed Gradient Boosted Regression Trees (GBRT) implemented via the XGBoost library. Before finalizing the model, we compared three candidate algorithms, namely Random Forest, Support Vector Regression, and XGBoost. A grid search procedure was used to identify the optimal set of hyperparameters for each model. Based on both predictive performance and interpretability considerations, XGBoost was selected as the final model because it provided an appropriate balance between coefficient of determination (R²) and root mean squared error (RMSE).”

In the limitation, we emphasized that:

“This study is based on data from China, and the results may not be directly generalizable to other countri

---

## [Decision Letter · Decision Letter 1]

30 Nov 2025

Effectiveness of Roadside Alcohol Testing in Reducing Fatal Accidents and Fatal Drinking-Driving Accidents: A Multi-City Study in China

PONE-D-25-37604R1

Dear Dr. Li,

We’re pleased to inform you that your manuscript has been judged scientifically suitable for publication and will be formally accepted for publication once it meets all outstanding technical requirements.

Kind regards,

Quan Yuan, Ph.D.

Academic Editor

PLOS ONE

Additional Editor Comments (optional):

Reviewers' comments:

Reviewer's Responses to Questions

**Comments to the Author**

Reviewer #1: All comments have been addressed

Reviewer #2: All comments have been addressed

2. Is the manuscript technically sound, and do the data support the conclusions?

Reviewer #1: Yes

Reviewer #2: Yes

3. Has the statistical analysis been performed appropriately and rigorously?

Reviewer #1: Yes

Reviewer #2: Yes

4. Have the authors made all data underlying the findings in their manuscript fully available?

Reviewer #1: Yes

Reviewer #2: Yes

5. Is the manuscript presented in an intelligible fashion and written in standard English?

Reviewer #1: Yes

Reviewer #2: Yes

Reviewer #1: Please double check the writing format of the article, especially if some of the references still contain relevant content written in Chinese. Does this meet the requirements of the journal? There are no further opinions on the content

Reviewer #2: All my comments have been addressed.

All my comments have been addressed.The manuscript can be accepted now.

**Do you want your identity to be public for this peer review?** For information about this choice, including consent withdrawal, please see our Privacy Policy

Reviewer #1: No

Reviewer #2: **Yes: ** Gen Li

---

## [Editor Report · Acceptance letter]

PONE-D-25-37604R1

PLOS One

Dear Dr. Li,

I'm pleased to inform you that your manuscript has been deemed suitable for publication in PLOS One. Congratulations! Your manuscript is now being handed over to our production team.

Kind regards,

on behalf of

Dr. Quan Yuan

Academic Editor

PLOS One